# p16 Expression in Multinucleated Stromal Cells of Fibroepithelial Polyps of the Anus (FEPA): A Comprehensive Review and Our Experience

**Milena Gulinac** [1,2]**, Tsvetelina Velikova** [2] **, Latchezar Tomov** [2,3,*] **and Dorian Dikov** [1,4]

[1] Department of General and Clinical Pathology, Medical University of Plovdiv, Bul. Vasil Aprilov 15A, 4000 Plovdiv, Bulgaria; mgulinac@hotmail.com (M.G.); dorian.dikov@hotmail.com (D.D.)

[2] Medical Faculty, Sofia University St. Kliment Ohridski, Kozyak 1 Str., 1407 Sofia, Bulgaria; tsvelikova@medfac.mu-sofia.bg

[3] Department of Informatics, New Bulgarian University, Montevideo 21 Str., 1618 Sofia, Bulgaria

[4] Department of General and Clinical Pathology, Grand Hospital de l'Este Francilien, Medical Faculty, 77600 Jossigny, France

**\*** Correspondence: lptomov@nbu.bg

**Abstract:** Fibroepithelial polyps of the anus (FEPA) are a common benign polypoid proliferation of the stroma covered by squamous epithelium. They are also an often-overlooked part of pathological practice. Currently, immunohistochemistry (IHC) for p16 is the only recommended test for anal intraepithelial neoplasia, but the expression of p16 in stromal multinucleated atypical cells in FEPA has not been described. We aimed to evaluate the expression of p16 in stromal multinucleated atypical cells in FEPA and its role as a diagnostic biomarker to determine the origin of the atypical multinucleated cells in the stroma of FEPA and to rule out the possibility of a neoplastic process. Therefore, we researched a series of 15 FEPA in middle-aged patients histologically and by IHC. Examination of the subepithelial connective tissue from the FEPA showed bizarre, multinucleated cells, while their causal relationship with human papillomavirus (HPV) infection was rejected. In all cases, these cells showed mild to moderate atypical nuclear features and positive expression for p16, while the overlying squamous epithelium was negative. We concluded that FEPA are benign lesions in the stroma where mononuclear and multinucleated (sometimes atypical) cells showing fibroblastic and myofibroblastic differentiation can be found. Nevertheless, we believe that these cells have a practical diagnostic significance, although sometimes the presence of giant cells is difficult to establish, especially in the inflammatory context. The histological similarity between FEPA and normal anal mucosa supports the hypothesis that FEPA may represent the reactive hyperplasia of subepithelial fibrous connective tissue of the anal mucosa.

**Keywords:** fibroepithelial polyp; anus; p16; immunohistochemistry (IHC)

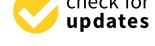



## 1. Introduction

Fibroepithelial polyps are quite common benign polypoid proliferations of myxoid or collagenous stroma, mostly of epithelial origin, covered by squamous epithelium [1,2]. In addition to the anal area, fibroepithelial polyps can also be found in the vulva, vagina, and female urethra [3,4]. Fibroepithelial polyps of the anus (FEPA) are an often-overlooked part of pathological practice. Therefore, comprehensive studies and analyses are essential to elucidate their clinical significance, molecular characteristics, and potential associations with other conditions. There is a need for standardized diagnostic criteria, improved awareness among healthcare professionals, and continued research to enhance our understanding of FEPA's etiology, clinical behavior, and optimal management strategies. By addressing these aspects, we can contribute to better patient care and refined diagnostic approaches and potentially identify novel therapeutic targets for this less-explored condition [5].

In line with this, finding proper biomarkers and unraveling the molecular signatures associated with FEPA could significantly enhance early diagnosis, prognostication, and personalized treatment strategies [5]. A suitable biomarker may pave the way for targeted therapies and interventions tailored to the specific characteristics of individual cases, ultimately improving patient outcomes and contributing to the advancement of precision medicine in colorectal pathology. However, there is still no evidence for suitable biomarkers for these purposes.

In our review, we aim to analyze comprehensively the current state of knowledge regarding FEPA. This includes a detailed exploration of their clinical characteristics, histopathological features, and potential molecular markers. Furthermore, we endeavor to critically evaluate existing diagnostic and therapeutic approaches while highlighting emerging trends and future directions in FEPA research. Through this synthesis of information, our goal is to contribute to a deeper understanding of FEPA, fostering advancements in the clinical management and scientific investigation of this intriguing pathological entity. Additionally, we aim to share our clinical experience in dealing with FEPA cases since, currently, immunohistochemistry (IHC) for p16 is the only recommended test for anal intraepithelial neoplasia, but the expression of p16 in stromal multinucleated atypical cells in FEPA has not been described. We evaluated the expression of p16 in stromal multinucleated atypical cells in FEPA and its role as a diagnostic biomarker to determine the origin of the atypical multinucleated cells in the stroma of FEPA and to rule out the possibility of a neoplastic process.

## 2. Characteristics of Fibroepithelial Polyps of the Anus

FEPA are considered normal anatomic variations that arise from the base of Morgagni's rectal columns at the dentate line. Their diverse etiology is usually due to inflammatory, infectious, reactive, or neoplastic changes. The association between FEPA and chronic inflammation has also been supported by Papadopoulos et al. in their case report of a 38-year-old woman with an anal fistula [2].

FEPA present an intriguing interaction with chronic inflammation, contributing to the complexity of their pathogenesis. Chronic inflammation plays a pivotal role in the initiation, development, and progression of these polyps. Histologically, FEPA lesions often exhibit characteristics typical of chronic inflammatory processes, such as increased vascularity, infiltration of immune cells, and stromal fibrosis. The inflammatory infiltrate commonly includes lymphocytes and plasma cells, indicating a sustained immune response within the polyp microenvironment [6].

The chronic inflammatory microenvironment of FEPA is dynamic and involves intricate interactions between various immune cells and stromal components. The specific triggers initiating chronic inflammation in FEPA are not fully elucidated, but factors such as mechanical irritation, infectious agents, or local trauma have been proposed as potential contributors. Moreover, chronic inflammation contributes to the architectural changes observed in FEPA, including hypertrophy of the overlying epithelium and alterations in the stromal composition [2].

Understanding the role of chronic inflammation in FEPA is essential for clinicians and researchers alike. While a FEPA is generally considered a benign lesion, chronic inflammation remains a critical aspect to consider in the broader context of colorectal pathology [7]; it not only impacts the histopathological features of the polyps, but may also influence clinical presentation and patient outcomes. Identifying inflammatory markers within FEPA lesions, as well as molecular and cellular mechanisms driving chronic inflammation in FEPA, may offer diagnostic and prognostic insights. Additionally, exploring the immune response in FEPA could unveil novel therapeutic avenues, potentially targeting the inflammatory pathways implicated in its pathogenesis.

In addition, Fellegara et al. reported an interesting case report of a FEPA harboring morphological evidence of epithelial vascular pseudoinvasion because of mechanical traumatism, such as a previous biopsy or stool passage as in their case, which could

play an essential role in epithelial endovascular displacement [8]. Histologically, they contain oedematous stroma, chronic inflammatory cells, mast cells and mononucleated and multinucleated stromal cells with fibroblastic and myofibroblastic differentiation, and hyperplastic squamous cell epithelium [1,2].

Furthermore, it is essential to point out that FEPA are a quite common finding, established in about 45% of patients undergoing proctoscopy, often affecting males with a Male:Female ratio of 2:1 [9,10]. Moreover, the most common age at which FEPA are diagnosed is middle to late adulthood, but sporadic cases have been reported in adolescent patients [10].

The summarized data in the literature give grounds to assume that FEPA are benign lesions of the anal canal considered to originate from the anal papillae, which are projections at the mucocutaneous junction of the upper part of the anal canal [2]. They can undergo hypertrophy into the rectum after repeated inflammatory episodes and be confused with adenomatous polyps or hemorrhoids. However, from a morphological perspective, FEPA are small protrusions (2–5 mm) that seldom exceed 2 cm [9]. Hence, big FEPA with a diameter of more than 3 cm should be treated carefully because of the elevated risk of misdiagnosis. From an endoscopic point of view, FEPA have several distinctive characteristics. Firstly, their mucosa is whitish, which could help to distinguish them from the reddish appearance of an adenomatous lesion. Secondly, the FEPA's stalk originates from the squamous side of the dentate line. Lastly, biopsy forceps or the cold polypectomy closure are more painful to FEPA compared to adenomas [11].

Furthermore, clinically, FEPA are symptomatic when prolapsing during defecation and occasionally necessitate digital repositioning, which strongly resembles rectal prolapse. The most common symptoms of this medical condition are pruritus, foreign body sensation, mucus discharge, a sense of incomplete evacuation, and discomfort while sitting [12]. A smooth mass inside the anal verge should be considered a FEPA, especially in patients with chronic anal irritation or locoregional infection history. In persons with poor anal hygiene, local disease, or bowel dysfunction, these lesions could even become inflamed and edematous with the appearance of hemorrhoids but with a lack of thick-walled veins or any evidence of bleeding and organizing thrombi [13].

According to numerous published scientific papers on the subject, FEPA may be reactive hyperplasia of the subepithelial connective tissue of the anal mucosa and the presence of mononuclear and multinucleated (sometimes atypical) stromal cells showing fibroblast and myofibroblast differentiation. This supports the hypothesis that cells of chronic inflammation, including mast cells, through their fibrinolytic, fibrogenic, and angiogenic activity, may play an essential role in the pathogenesis of FEPA because the establishment of multinucleated cells in the stroma of FEPA is not a rare finding [1,9,14,15]. Their IHC status has been studied in detail and described in many scientific publications. They prove that multinucleated giant cells in the stroma of fibroepithelial polyps are positive for vimentin, desmin (in 30%) [1,16], CD34 [9], and a small percent are negative for smooth muscle actin (SMA) [1,16]. However, to our knowledge, the expression of p16 only in stromal multinucleated atypical cells in FEPA has not been described.

Although the most common reasons for the development of fibroepithelial polyps of the anus are chronic inflammation and irritation, following the principles of good medical practice, it is still essential to exclude intraepithelial neoplasia caused by human papillomavirus (HPV). For that purpose, as well as for categorizing borderline low-grade squamous intraepithelial lesions (LSIL)/high-grade squamous intraepithelial lesions (HSIL), the most useful approach is to perform an IHC assay with a p16 antibody. Furthermore, IHC positivity is commonly considered a surrogate marker for oncogenic HPV infection. A strong positive expression of the squamous epithelium is a marker for high-risk HPV and indicates HSIL [16–18]. Inactivating the retinoblastoma tumor suppressor protein (Rb) by the viral E7 oncoprotein following viral integration into the host genome leads to overexpression of p16 [19].

As mentioned before, the differential diagnosis between FEPA and other anal pathology is of paramount importance, especially in those FEPA with atypical macroscopic features such as enormous size, ulceration, or the presence of a cutaneous horn. These FEPA could mask malignant tumors, including leiomyosarcoma, anorectal carcinoma, malignant lymphoma, and verrucous carcinoma, especially in a patient with a history of infection with a low or high risk of HPV genotypes and chronic anal irritation [16,20], as reported in a case by Mercer et al. [21]. Thus, all available diagnostic tools should be used in order not to misdiagnose the condition.

The molecular signatures of FEPA are still far from coming into focus, although such investigations would be shedding light on the intricate mechanisms underlying their development and progression [22]. While FEPA are primarily characterized by known architectural and histological features, no molecular studies have revealed distinct genetic alterations and signaling pathways associated with these lesions. However, molecular analyses would suggest whether FEPA exhibit alterations in genes involved in cell cycle regulation, stromal–epithelial interactions, and immune response modulation, but, to date, we have no such data [22].

On the other hand, IHC studies have provided insights into the molecular landscape of FEPA, highlighting altered expression patterns of certain proteins associated with cellular proliferation, apoptosis, and inflammation [22]. The investigation of FEPA contributes to our understanding of their pathophysiology and offers potential diagnostic markers and therapeutic targets. Continued research in this realm promises to uncover molecular markers that can aid in the accurate diagnosis, risk stratification, and management of patients with FEPA. As for therapeutic management, there is scarce literature data, but a few published articles recommend complete removal and excision with an electrothermal bipolar vessel sealing system, electrocauterization, or ultrasonic energy [23].

### 3. p16 Expression in Fibroepithelial Polyps of the Anus

p16 is a tumor suppressor gene essential for cell proliferation that acts as a cyclin-dependent kinase inhibitor 2A (CDKN2A) to cause cell cycle arrest [24], also known as multiple tumor suppressor 1 (MTS1) and p16 INK4a. Additionally, high levels of p16 protein production may suppress cell proliferation by preventing the cell cycle from progressing past the G1/S restriction point [25]. Literature data have shown that point mutation, promoter hypermethylation, or homozygous deletion can cause the inactivation of p16, a common finding in many human malignancies [26]. Oncology patients frequently have genetic variations in the p16 gene. A key strategy for mutating tumor suppressor genes is methylation [27]. For instance, the methylation status of the p16 gene promoter has been extensively researched. It has been shown to be highly related to the onset of numerous malignancies, such as bladder cancer, lung cancer, brain cancer, and esophageal cancer. They have been associated with the onset, progression, and prognosis of aberrant methylation of the promoters of the p16 gene [28]. Similarly, a study by Veganzones-de-Castro et al. raised the possibility that the clinicopathologic characteristics of CRC may be significantly correlated with promoter hypermethylation of the p16 gene [29].

Furthermore, p16 has been consistently linked to premalignant lesions in current clinical practice. It has been successfully utilized to predict the progression of Barrett's esophagus and cervical biopsies to high-grade dysplasia or carcinoma [30]. The diagnostic accuracy of p16 has been reported in anal intraepithelial neoplasia (AIN) in patients with suspected HPV infection, especially those with high-grade AIN. Positive p16 expression was characterized by diffuse moderate-to-strong cytoplasmic and nuclear staining [20]. A similar IHC profile with strong p16 expression was also described in the stratified mucin-producing intraepithelial anal lesion. However, there is scarce data on the p16 expression pattern in patients with FEPA.

p16 is one of the most direct links between cell-cycle control and cancer. p16 inactivates cyclin-dependent kinases that phosphorylate Rb; p16 can decelerate the cell cycle. Rb phosphorylation status, in turn, influences the expression of p16. In

HPV infection, the HPV oncogenes E6 and E7 can inactivate phosphorylated Rb (pRB) and thus lead to p16 overexpression, most commonly studied and used in evaluating anogenital lesions (cervical, vulvar, vaginal, anal, penile). Also, the examination of this protein is useful in the evaluation of head and neck carcinoma and different gynecologic tract tumors; p16 overexpression is more frequently seen in high-grade endometrial and ovarian carcinoma (serous, clear cell, and high-grade endometrioid) compared to low-grade tumors [31,32].

According to literature data, the loss of p16 protein expression often increases in aging or senescent cells, which eventually drives cell death and apoptosis [17,18]. In contrast to the other aging biomarkers mentioned above, p16 is directly involved in the onset and maintenance of senescence. Unbiased recent findings from a genome-wide association analysis linked p16 to human aging and disorders associated with old age [33]. In line with this, measuring p16 in peripheral blood could be a promising and reliable marker for senescence. Furthermore, p16-based molecular age has been validated in clinical studies, including in cancer survivors—children, adolescents, and young adults [34]. Furthermore, rejuvenation of the aged brain immune cells in mice was studied in regards to using p16 as a marker for aging [35].

Studies on various chemokines, interleukins, and other molecules suggested that p16 regulates the immunological surveillance in tissues. In short, suppressing p16 in tumor cells reduces the expression of such immune mediators, which may change the tumor microenvironment and impair immunological surveillance [36]. In turn, this may remodel the tumor microenvironment, thereby impairing immunological surveillance [36].

The expression of p16—the marker of cellular aging and degeneration—supports Epstein et al.'s theory for a degenerative cellular phenomenon [18]. Similarly, LaPak and colleagues pointed out several pieces of evidence that p16 is a biomarker of aging and a cause of aging in many cell types. Their data also support the hypothesis that p16 expression denotes cellular aging and, at the same time, leads to a decrease in regenerative potential [37]. From a pathogenetic point of view, giant multinucleated stromal cells in FEPA represent a degenerative cellular phenomenon reflecting chronic mucosal irritation and inflammation.

However, the expression of the p16 suppressor gene was not investigated in the context of FEPA. Since p16 plays a pivotal role in regulating the cell cycle by inhibiting the activity of cyclin-dependent kinases, studies have indicated alterations in the expression of p16 in various neoplastic conditions, including those affecting the anorectal region. It is possible that aberrant p16 expression patterns in FEPA may signify early disruptions in cell cycle control mechanisms, contributing to the pathophysiology of the condition.

## 4. Our Experience: A Comprehensive Analysis of 15 Fibroepithelial Polyps of the Anus Cases

We hypothesized that aberrant p16 expression may potentially contribute to the pathogenesis of FEPA, especially in the scarce data on p16 expression pattern in patients with this condition. Thus, our study sought to determine the expression of p16 in stromal multinucleated atypical cells in FEPA and its role as a diagnostic biomarker. We also aimed to determine the origin of the atypical multinucleated cells in the stroma of FEPA and to rule out the possibility of a neoplastic process or HPV infection. To our knowledge, no studies have explored the expression of p16 in FEPA. Therefore, we researched a series of 15 consecutive FEPA specimens in middle-aged patients (10 men and 5 women) histologically by light microscopic and IHC. All patients came with a complaint of polypoid formation in the anal area.

The 15 cases were identified from the Grand Hospital de l'Este Francilien, Jossigny, France, and the Departments of General and Clinical Pathology of St. George University Hospital of Plovdiv, Bulgaria. Two pathologists histologically evaluated all 15 cases: 1. Chief Assistant Dr. Milena Gulinac, Ph.D. and 2. Associate Professor Dr. Dorian Dikov, Ph.D. Clinical data and macroscopic details were obtained from surgical pathology reports.

The age of the patients ranged from 45 to 68 years (mean 61.4 years). Clinically, four of the patients were asymptomatic, while the remaining patients described anal pain and anal discomfort for a median period of 9.3 months. In only three patients, a fibroepithelial polyp was associated with hemorrhoids. The patients did not have severe comorbidities, but half of them presented with arterial hypertension and poor lipid profile. None of the patients were immunized with the HPV vaccine.

Histology analysis of the tissues was performed using the automatic tissue processor "DIAPATH EN ISO 9001:2000", and 4–5 µm formalin-fixed paraffin-embedded tissue (FFPE) sections underwent routine staining with hematoxylin-eosin (HE) to determine the presence of individual histological features. There was only one criterion for immunohistochemical examination with p16 in these 15 cases, and that was the presence of atypical giant multinucleated cells in the stroma of the fibroepithelial polyps, established histologically on conventional staining with HE. In fact, they represented individual histological features because they are a rare finding in this type of pathology.

We used P16-INK4A antibodies to detect the expression of the p16 gene in formalin-fixed, paraffin-embedded tissue, using a visualization system. Usually, tonsils are used for positive and negative tissue controls. Additionally, scattered reticulated cryptepithelial cells must demonstrate a moderate to strong nuclear and cytoplasmic staining reaction. Dispersed germinal center macrophages/dendritic cells must show at least a weak but distinct nuclear and cytoplasmic staining reaction. Finally, lymphocytes and normal superficial squamous epithelial cells should not demonstrate positive staining. Positive staining is defined as strong nuclear and cytoplasmic expression in all stromal atypical multinucleated cells. Cytoplasmic-only staining, diffuse blush, and focal/patchy staining were considered negative.

In its daily routine, the pathology department uses a monthly control tissue, most commonly tonsil, for the purpose of positive and negative control for more than one antibody, and the head of the department reports the results. For this reason, imaging positive and negative control tissue for academic development purposes is not routinely used except in large-scale research. In addition, the described 15 cases and all biopsies received in the department were part of the routine biopsy work of pathologies in this author's collective. This necessarily includes preliminary validation of the antibodies used in the study.

IHC staining for p16 expression was performed in all 15 cases. Strong positive p16 expression in giant multinucleated stromal cells was determined in all cases (100%) (Figure 1A,B).

IHC examination of p16 showed strong positive nuclear and cytoplasmic expression in all stromal atypical multinucleated cells. However, none of the cases showed positive expression of p16 in the lining squamous epithelium, firmly ruling out the presence of HPV infection. The lack of p16 expression in the lining epithelium was unsurprising because no morphological changes comparable to HPV infection were observed.

Positive expression is not detected in the lining stratified squamous epithelium because cytoplasmic-only staining with weak intensity and other focal/patchy patterns should be considered negative. As a rule, positive staining is demonstrated as "block" staining. The latter means strong nuclear and cytoplasmic expression in a continuous segment of cells (at least 10–20 cells). However, block positivity in the squamous epithelium should involve basal and parabasal layers—this type of staining is not visualized in these cases.

Our study initially focused on rejecting or confirming the infectious etiology of the case studies using p16 because it is most helpful in evaluating anogenital lesions. However, after IHC was performed, we found a strong expression of mild to moderately atypical giant multinucleated cells in the stroma of the polyps and the absence of squamous epithelium in all 15 cases.

The results we obtained favor p16-mediated, replication-induced cellular senescence in the conditions of fibro-epithelial tissue proliferation, as it is in FEPA [38]. Additionally, p16 IHC is not recommended as a routine adjunct assessment when the biopsy interpretation

is negative; any identified p16-positive area must meet H&E morphologic criteria for a high-grade lesion, and only then could it be reinterpreted as such [39].

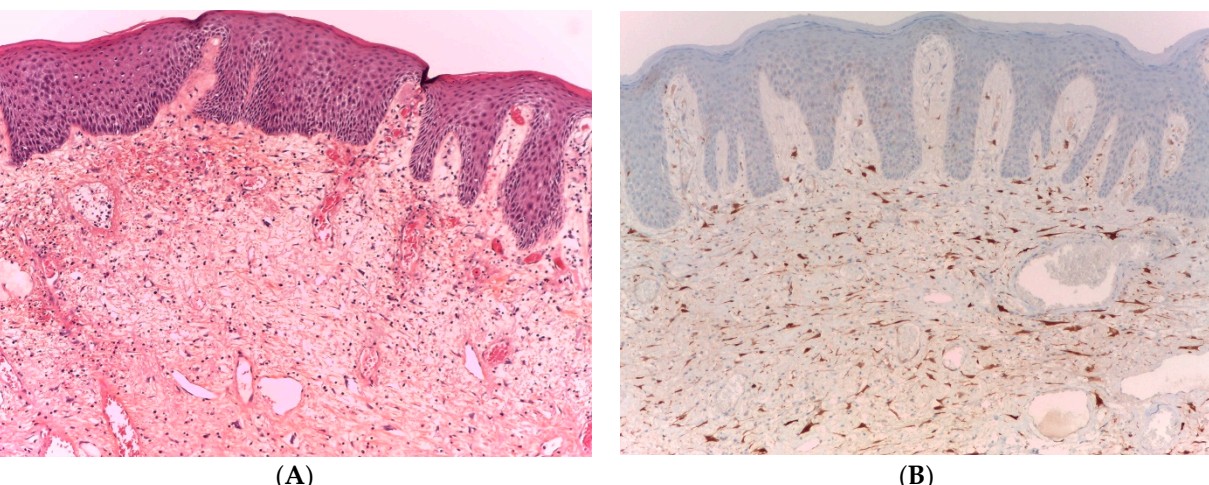

**(A)**                                                            **(B)**

**Figure 1.** FEPA, HE (100× magnification). (**A**) The fibroepithelial polyp comprises a collagenous stroma covered by a squamous epithelium microphotograph. The FEPA contain mononucleated and multinucleated stromal cells with fibroblastic and myofibroblastic differentiation and mast cells, as shown by the arrows in the histology of the specimen; (**B**) FEPA, p16 +. Strong nuclear and cytoplasmic expression of p16 in giant multinucleated stromal cells with negative expression in the overlying squamous epithelium.

Taken together, we aimed to determine the origin of the atypical multinucleated cells in the stroma of fibroepithelial polyps and to rule out the possibility of a neoplastic process while rejecting the causal relationship between their presence in anal fibroepithelial polyps and HPV infection. When we researched (histologically and using IHC) a series of 15 FEPA in middle-aged patients by examination of the subepithelial connective tissue from the FEPA, we discovered bizarre, multinucleated cells. In all cases, these cells showed mild to moderate atypical nuclear features and positive expression for p16, while the overlying squamous epithelium was negative.

Our investigation has some limitations. First, the relatively small number of case studies allows for some conclusions to be made, but large-scale studies are needed to confirm these results. Secondly, we did not perform additional analysis of the FEPA; however, it is worth establishing some associations with the clinical picture and also assessing molecular alterations via genetics.

Currently, IHC for p16 is the only recommended test for anal intraepithelial neoplasia, but the expression of p16 in stromal multinucleated atypical cells in FEPA has not been described. We believe this has a practical diagnostic significance, and, sometimes, the presence of giant cells is difficult to establish, especially in the inflammatory context. Indeed, p16 is tested immunohistochemically in anal and other lesions in routine practice. In other words, p16 is routinely tested to rule out an HPV-associated lesion in the overlying squamous epithelium. In our study, we initially focused on rejecting or confirming the infectious etiology of the case studies using p16 because it is most useful for the evaluation of anogenital lesions. However, after IHC was performed, we found a strong expression of mild-to-moderately atypical giant multinucleated cells in the stroma of the polyps and the absence of it in squamous epithelium in all 15 cases. The immunohistochemical status of these multinucleated cells has been extensively studied and described in many scientific publications. They demonstrate that multinucleated giant cells in the stroma of fibroepithelial polyps are positive for vimentin, desmin, CD34, and, in a small percentage, SMA, but p16 expression only in stromal multinucleated atypical cells in FEPA is not described. The expression of p16—the marker of cellular aging and degeneration—supports

the theory of a degenerative cellular phenomenon. Many data also support the hypothesis that p16 expression denotes cellular aging and, at the same time, leads to a decrease in regenerative potential.

We concluded that FEPA are benign lesions in the stroma where mononuclear and multinucleated (sometimes atypical) cells showing fibroblastic and myofibroblastic differentiation can be found. The histological similarity between FEPA and normal anal mucosa supports the hypothesis that FEPA may represent reactive hyperplasia of subepithelial fibrous connective tissue of the anal mucosa.

## 5. Conclusions

In conclusion, our results and those of other authors confirm the expression of p16 in stromal multinucleated atypical cells in FEPA, a phenomenon that has not yet been described fully. Moreover, the bizarre, multinucleated cells observed in the subepithelial connective tissue from FEPA showed mild-to-moderate atypical nuclear features in addition to positive expression for p16. Although IHC for p16 is the only recommended test for anal intraepithelial neoplasia, p16 expression has a practical diagnostic significance, particularly in scenarios where establishing the presence of giant cells is challenging within an inflammatory context. These findings contribute to a deeper understanding of the histopathological features of FEPA, emphasizing the potential utility of p16 expression as a valuable diagnostic marker in challenging clinical scenarios. Further research exploring the clinical implications and long-term outcomes associated with p16-positive FEPA cases is warranted to enhance our diagnostic and prognostic capabilities in anal pathology.

**Author Contributions:** Conceptualization, M.G. and D.D.; Data curation, M.G.; Formal analysis, T.V. and L.T.; Funding acquisition, M.G.; Investigation, M.G. and D.D.; Methodology, M.G.; Project administration, D.D.; Resources, M.G. and D.D.; Software, L.T.; Supervision, D.D. and T.V.; Validation, M.G., D.D. and T.V.; Visualization, T.V.; Writing—original draft, M.G. and T.V.; Writing—review editing, D.D. and T.V. All authors have read and agreed to the published version of the manuscript.

**Funding:** This study is financed by the European Union-NextGenerationEU, through the National Recovery and Resilience Plan of the Republic of Bulgaria, project No. BG-RRP-2.004-0008.

**Institutional Review Board Statement:** The study of the 15 FEPA cases was conducted in accordance with the Declaration of Helsinki and approved by the Ethics Committee of the University of Plovdiv, Department of General and Clinical Pathology, with protocol No. 67/16.12.2022.

**Informed Consent Statement:** Informed consent was obtained from all subjects involved in the study.

**Acknowledgments:** This study is financed by the European Union-NextGenerationEU, through the National Recovery and Resilience Plan of the Republic of Bulgaria, project No. BG-RRP-2.004-0008.

**Conflicts of Interest:** All authors declare no conflicts of interest.

## Abbreviations

| | |
|---|---|
| AIN | Anal intraepithelial neoplasia |
| CD34 | Cluster of differentiation 34 |
| CDKN2A | Cyclin-dependent kinase inhibitor 2A |
| FEPA | Fibroepithelial polyps of the anus |
| FFPE | Formalin-fixed paraffin-embedded tissue |
| HE | Hematoxylin-eosin |
| HPV | Human papillomavirus |
| IHC | immunohistochemistry |
| HSIL | High-grade squamous intraepithelial lesions |
| LSIL | Low-grade squamous intraepithelial lesions |
| MTS1 | Multiple tumor suppressor 1 |
| pRb | Phosphorylated Rb |
| Rb | Retinoblastoma |
| SMA | Smooth muscle actin |

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
