# Peer review of "p16 Expression in Multinucleated Stromal Cells of Fibroepithelial Polyps of the Anus (FEPA): A Comprehensive Review and Our Experience"

_gastroent, doi:10.3390/gastroent15020029_

Round 1

Reviewer 1 Report

Comments and Suggestions for Authors

In this work the authors describe the presence of multinucleated cells expressing p16 in the histological preparations of fibroepithelial polyps of the anus from 15 patients.

The paper is of interest, however the exposition should be more concise, avoiding
often repetitive considerations. I recommend adding a table with the abbreviations
used in the text.

Author Response

Report 1

Comments and Suggestions for Authors

In this work the authors describe the presence of multinucleated cells expressing p16 in the histological preparations of fibroepithelial polyps of the anus from 15 patients.

The paper is of interest, however the exposition should be more concise, avoiding often repetitive considerations. I recommend adding a table with the abbreviations used in the text.

  • Authors` responses: Thank you for taking the time to review our paper and for providing valuable feedback. We sincerely appreciate your comments and suggestions, which we believe will improve the clarity and impact of our work.
  • We agree that the exposition of our findings can be more concise, and we have revised the manuscript accordingly to avoid repetitive considerations. Additionally, we have included a table with the abbreviations used in the text (at the end of the manuscript), which we believe will enhance the readability of the paper for our readers.
  • Once again, we are grateful for your insightful comments and for helping us to strengthen our manuscript. Please find the revised version attached for your review.

Reviewer 2 Report

Comments and Suggestions for Authors

This review and experience by Milena Gulinac et al. analyzed the current state of knowledge regarding clinical characteristics, histopathological features, and potential molecular markers for Fibroepithelial polyps of the anus (FEPA). The authors shared their clinical experience in dealing with FEPA cases. The authors note the expression of p16 in stromal multinucleated atypical cells in FEPA, a phenomenon that has not yet been described. The authors conclude that further research exploring the clinical implications and long-term outcomes associated with p16-positive FEPA cases is warranted to enhance our diagnostic and prognostic capabilities in anal pathology. The topic is valid, and the work is scientifically sound. The study methods are appropriate. The manuscript is well written overall and adequate references are included. The manuscript can be improved by addressing the following concerns.

-       The authors may want to rewrite the Abstract section to clearly describe the aim of this review. It does not read well and difficult to follow unlike the Introduction section.

-       Consider including more clinical details of 15 FEPA cases, like duration of symptoms, presence of comorbidities, HPV vaccination status and other factors that could have influenced the study results.

-       Add limitations of this study in the Discussion section.

Author Response

Report 2

Comments and Suggestions for Authors

This review and experience by Milena Gulinac et al. analyzed the current state of knowledge regarding clinical characteristics, histopathological features, and potential molecular markers for Fibroepithelial polyps of the anus (FEPA). The authors shared their clinical experience in dealing with FEPA cases. The authors note the expression of p16 in stromal multinucleated atypical cells in FEPA, a phenomenon that has not yet been described. The authors conclude that further research exploring the clinical implications and long-term outcomes associated with p16-positive FEPA cases is warranted to enhance our diagnostic and prognostic capabilities in anal pathology. The topic is valid, and the work is scientifically sound. The study methods are appropriate. The manuscript is well written overall and adequate references are included. The manuscript can be improved by addressing the following concerns.

  • Authors` responses: Thank you very much for your thorough review of our paper. We are grateful for your positive comments regarding the validity and scientific soundness of our work, as well as your constructive suggestions for improvement.

-       The authors may want to rewrite the Abstract section to clearly describe the aim of this review. It does not read well and difficult to follow unlike the Introduction section.

  • Authors` responses: We acknowledge your feedback regarding the Abstract section, and we have revised it to better clarify the aim of our review and enhance readability.

-       Consider including more clinical details of 15 FEPA cases, like duration of symptoms, presence of comorbidities, HPV vaccination status and other factors that could have influenced the study results.

  • Authors` responses: Additionally, we have included more clinical details of the 15 FEPA cases in our study, when applicable and had data.  

-       Add limitations of this study in the Discussion section.

  • Authors` responses: Thank you for the constructive suggestion, we have added a section in the Discussion highlighting the limitations of our study. We believe that addressing these limitations contributes to a more comprehensive understanding of our findings and their implications.
  • Thank you once again for your valuable input, which has undoubtedly strengthened our manuscript. We look forward to your feedback on the revised version.

Reviewer 3 Report

Comments and Suggestions for Authors

Considering that p16 is a marker of cell proliferation, please make some more comments about the practical points of your study. Considering that lesions initially classified as anal intraepithelial neoplasia (AIN) 2 that are p16-negative are downgraded to LSIL and lesions initially classified as AIN 2 that are p16-positive are upgraded to HSIL, clarify further the practical usefulness of your study. 

Comments on the Quality of English Language

Please read carefully the text and make some adjustments to simplify some paragraphs. Example: The first two sentences of the introduction could be simplified in one sentence.

Author Response

Report 3

Comments and Suggestions for Authors

Considering that p16 is a marker of cell proliferation, please make some more comments about the practical points of your study. Considering that lesions initially classified as anal intraepithelial neoplasia (AIN) 2 that are p16-negative are downgraded to LSIL and lesions initially classified as AIN 2 that are p16-positive are upgraded to HSIL, clarify further the practical usefulness of your study.

  • Authors` responses: Thank you for your insightful comments and suggestions regarding the practical implications of our study and the quality of the English language in our manuscript. We appreciate your attention to detail and are committed to improving the clarity and impact of our work.
  • Regarding the practical points of our study, we are thankful for the great question. We will start by saying that you are correct about the purpose for which p16 is tested immunohistochemically, in anal and other lesions, in routine practice, in other words p16 is routinely tested to rule out an HPV-associated lesion in the overlying squamous epithelium. In our study, we initially focused on rejecting or confirming the infectious etiology of the case studies, using p16, because it is most useful for the evaluation of anogenital lesions. However, after immunochemistry was performed, we found a strong expression of mild to moderately atypical giant multinucleated cells in the stroma of the polyps, and absence of it in squamous epithelium in all 15 cases. The immunohistochemical status оf these multinucleated cells has been extensively studied and described in many scientific publications. They demonstrate that multinucleated giant cells in the stroma of fibroepithelial polyps are positive for vimentin, desmin CD34 and in a small percentage negative for smooth muscle actin (SMA), but p16 expression only in stromal multinucleated atypical cells in FEPA is not described. The expression of p16 - the marker of cellular aging and degeneration, support theory for a degenerative cellular phenomenon. Many data’s also support the hypothesis that p16 expression denotes cellular aging and in the same time lead to decrease in regenerative potential.
  • We will further elaborate on these practical implications in our manuscript to provide a clearer understanding of the clinical relevance of our findings.

Comments on the Quality of English Language

Please read carefully the text and make some adjustments to simplify some paragraphs. Example: The first two sentences of the introduction could be simplified in one sentence.

  • Authors` responses: In terms of the English language quality, we carefully reviewed the text and adjusted paragraphs where necessary, to improve readability and flow.
  • Thank you again for your valuable feedback, which will undoubtedly enhance the overall quality of our manuscript. We look forward to incorporating these improvements and providing a revised version for your review.